# ON INDUCTIVE BIASES IN
# DEEP REINFORCEMENT LEARNING

## ABSTRACT

Many deep reinforcement learning algorithms contain inductive biases that sculpt the agent's objective and its interface to the environment. These inductive biases can take many forms, including domain knowledge and pretuned hyper-parameters. In general, there is a trade-off between generality and performance when we use such biases. Stronger biases can lead to faster learning, but weaker biases can potentially lead to more general algorithms that work on a wider class of problems. This trade-off is relevant because these inductive biases are not free; substantial effort may be required to obtain relevant domain knowledge or to tune hyper-parameters effectively. In this paper, we re-examine several domain-specific components that modify the agent's objective and environmental interface. We investigated whether the performance deteriorates when all these fixed components are replaced with adaptive solutions from the literature. In our experiments, performance sometimes decreased with the adaptive components, as one might expect when comparing to components crafted for the domain, but sometimes the adaptive components performed better. We then investigated the main benefit of having fewer domain-specific components, by comparing the learning performance of the two systems on a different set of continuous control problems, without additional tuning of either system. As hypothesized, the system with adaptive components performed better on many of the tasks.

The deep reinforcement learning (RL) community has demonstrated that well-tuned deep RL algorithms can master a wide range of challenging tasks. Human-level performance has been approached or surpassed on board-games such as Go and Chess (Silver et al., 2017), video-games such as Atari (Mnih et al., 2015; Hessel et al., 2018a; Horgan et al., 2018), and custom 3D navigation tasks (Johnson et al., 2016; Kempka et al., 2016; Beattie et al., 2016; Mnih et al., 2016; Jaderberg et al., 2016; Espeholt et al., 2018). These results are a testament to the generality of the overall approach. At times, however, the excitement for the constant stream of new domains being mastered by suitable RL algorithms may have over-shadowed the dependency on inductive biases of these agents, and the amount of tuning that is often required for these to perform effectively in new domains. A clear example of the benefits of generality is the AlphaZero algorithm (Silver et al., 2017). AlphaZero differs from the earlier AlphaGo algorithm (Silver et al., 2016) by removing all dependencies on Go-specific inductive biases and human data. After removing these biases, AlphaZero could not just surpass human performance on Go but also learn to play Chess and Shogi.

In general, there is a trade off between generality and performance when we inject inductive biases into our algorithms. Inductive biases take many forms, including domain knowledge and pretuned learning parameters. If applied carefully, such biases can lead to faster and better learning. On the other hand, fewer biases can potentially lead to more general algorithms that work out of the box on a wider class of problems. Crucially, most inductive biases are not free: for instance, substantial effort can be required to attain the relevant domain knowledge or pretune parameters. This cost is often hidden—for instance, one might use hyperparameters established as good in prior work on the same domain, without knowing how much data or time was spent optimising these, nor how specific the settings are to the given domain. Systematic studies about the impact of different inductive biases are rare and the generality of these different biases is often unclear. Another consideration is that inductive biases may mask the generality of other parts of the system as a whole; if a learning algorithm tuned for a specific domain does *not* generalize out of the box to a new domain, it can

be unclear whether it is due to the having the wrong inductive biases or whether the underpinning learning algorithm is lacking something important.

In this paper, we consider several commonly used domain heuristics, and investigate if (and by how much) performance deteriorates when we replace these with more general adaptive components, and we assess the impact of such replacements on the generality of the agent. We consider two broad ways of injecting inductive biases in RL agents: 1) sculpting the agent's objective (e.g., clipping and discounting rewards), 2) sculpting the agent-environment interface (e.g., repeating each selected action a hard-coded fixed number of times, or crafting of the learning agent's observation). Our contributions show that all the carefully crafted heuristics commonly used in Atari, and often considered essential for good performance on this domain, can be safely replaced with adaptive components, while preserving competitive performance across the benchmark. Furthermore, we show that this results in increased generality for an actor critic agent; the resulting fully adaptive system can be applied with no additional tuning on a separate suite of continuous control tasks that have different environmental interfaces. Here the adaptive system achieved much higher performance than a comparable system using the Atari tuned heuristics, and also higher performance than an actor-critic agent that was tuned for this benchmark (Tassa et al., 2018).

## 1 BACKGROUND

Reinforcement learning (Sutton & Barto, 1998) is a framework for learning and decision making under uncertainty, where an *agent* interacts with its *environment*, receiving an *observation* $O_t$, selecting an action $A_t$, and then receiving the next *reward* $R_{t+1}$, for each discrete time $t$.

**Problem setting:** The behaviour of an agent is formally specified by a policy $\pi(A_t|H_t)$: a probability distribution over actions conditional on all previous observations, i.e. on the agent's *history* $H_t = O_{1:t}$. The agent's objective is to find a policy that collects as much reward as possible, in each episode of experience. Crucially, it must learn such a policy without direct supervision, by trial and error. The amount of reward collected from time $t$ onwards - the *return* - is a random variable

$$G_t = \sum_{k=0}^{T_{end}} \gamma^k R_{t+k+1}, \tag{1}$$

where $T_{end}$ is the number of steps until episode termination and $\gamma \in [0, 1]$ is a constant discount factor (e.g., modelling monetary inflation, or a fixed additional probability of termination). The agent seeks an *optimal* policy, that maximizes *values* $v(H_t) = E_\pi[G_t|H_t]$. In *fully observable* environments the optimal policy depends on the last observation alone: $\pi^*(A_t|H_t) = \pi^*(A_t|O_t)$. Otherwise, the history may be summarized in an *agent state* $S_t = f(H_t)$. The agent's objective is then to *jointly* learn the state representation $f$ and policy $\pi(A_t|S_t)$, so as to maximize values. The fully observable case is formalized as a Markov Decision Process (Bellman, 1957).

**Actor-critic algorithms:** In the literature, two families of RL algorithms have particular prominence. *Value-based* algorithms learn to estimate values $v_\mathbf{w}(s)$ that approximate to the true values, $v_\mathbf{w}(s) \approx v_\pi(s) \equiv E_\pi[G_t|S_t = s]$, under some policy $\pi$. Values can be learned efficiently by exploiting a recursive decomposition $v_\pi(s) = \mathbb{E}[R_{t+1} + \gamma v_\pi(S_{t+1})|S_t = s]$, known as *Bellman equation*, used in in temporal difference learning (Sutton, 1988) by sampling and making incremental updates:

$$\Delta\mathbf{w}_t = (R_{t+1} + \gamma v_\mathbf{w}(S_{t+1}) - v_\mathbf{w}(S_t))\nabla_\mathbf{w} v_\mathbf{w}(S_t). \tag{2}$$

In *policy-based* reinforcement learning algorithms, the policy $\pi_{\boldsymbol{\theta}}(A_t|S_t)$ is parameterized directly, and a stochastic gradient estimate can be derived to update the parameters in the direction of steepest ascent of the value (Williams, 1992; Sutton et al., 2000): for instance, according to the update

$$\Delta\boldsymbol{\theta}_t = G_t \nabla \log \pi_{\boldsymbol{\theta}}(A_t|S_t). \tag{3}$$

Value-based and policy-based methods are combined in *actor-critic* algorithms. If a state value estimate is available, the policy updates can be computed from incomplete episodes by using the truncated returns $G_t^{(n)} = \sum_{k=0}^{n-1} \gamma^k R_{t+k+1} + \gamma^n v_\mathbf{w}(S_t)$ that bootstrap on the value estimate at state $S_{t+n}$ according to $v_\mathbf{w}$. This can reduce the variance of the updates. The variance can be further reduced by using state values as a baseline in policy updates, as in the *advantage* actor-critic update:

$$\Delta\boldsymbol{\theta}_t = (G_t^{(n)} - v_\mathbf{w}(S_t))\nabla_{\boldsymbol{\theta}} \log \pi_{\boldsymbol{\theta}}(A_t|S_t). \tag{4}$$

## 2 INDUCTIVE BIASES AND LEARNED SOLUTIONS

We now describe a few commonly used heuristics within the Atari domain, together with the adaptive replacements that we investigated in our experiments.

### 2.1 SCULPTING THE AGENT'S OBJECTIVE

Many current deep RL agents do not directly optimize the true objective that they are evaluated against. In particular, RL agents in the Atari environments are typically tasked with optimizing a different handcrafted objective that incorporates biases to make learning simpler. We consider two additional ways of sculpting the agent's objective that are still commonly used: reward clipping, and the use of fixed discounting of future rewards by a factor different from the one used for evaluation.

In many deep RL algorithms, the magnitude of the updates scales linearly with the returns. This makes it difficult to train the same RL agent, with the same hyper-parameters, on multiple domains, because good settings for hyper-parameters such as the learning rate vary across tasks. One common solution is to clip the rewards to a fixed range (Mnih et al., 2015), for instance $[-1, 1]$. This clipping makes the magnitude of returns and updates more comparable across domains. However, this also radically changes the agent objective, e.g., if all non-zero rewards are larger than one, then this amounts to maximizing the frequency of positive rewards rather than their cumulative sums. This can simplify the learning problem, and, when it is a good proxy for the true objective, can result in good performance. In other tasks, however, clipping can result in sub-optimal policies because the objective that is optimized is ill-aligned with the true objective.

PopArt (van Hasselt et al., 2016; Hessel et al., 2018b) was introduced as a principled solution to learn effectively irrespective of the magnitude of returns. PopArt works by tracking the *mean* $\mu$ and *standard deviation* $\sigma$ of bootstrapped returns $G_t^{(n)}$. Temporal difference errors on value estimates can then be computed in a normalized space, with $n_\mathbf{w}(s)$ denoting the normalized value, while the unnormalized values (needed, for instance, for bootstrapping) are recovered by a linear transformation $v_\mathbf{w}(s) = \mu + \sigma * n_\mathbf{w}(s)$. Doing this naively increases the non-stationarity of learning since the unnormalized predictions for all states change every time we change the statistics. PopArt therefore combines the adaptive rescaling with an inverse transformation of the weights at the last layer of $n_\mathbf{w}(s)$, thereby preserving outputs precisely under any change in statistics $\mu \to \mu'$ and $\sigma \to \sigma'$. This is done *exactly* by updating weights and biases as $w' = w\sigma/\sigma'$ and $b' = (\sigma b + \mu - \mu')/\sigma'$.

Discounting is part of the traditional MDP formulation of RL. As such, it is often considered a property of the problem rather than a tunable parameter on the agent side. Indeed, sometimes, the environment does define a natural discounting of future rewards (e.g., inflation in a financial setting). However, even in episodic settings where the agent should maximize the undiscounted return, a constant discount factor is often used to simplify the learning problem (by having the agent focus on a relatively short time horizon). Optimizing this proxy of the true return often results in the agent achieving superior performance even in terms of the undiscounted return (Machado et al., 2017). This benefit comes with the cost of adding a hyperparameter, and performance can be quite sensitive to it—i.e., learning might be fast if the discount is small, but the solution may be too myopic.

Instead of tuning the discount manually, we use meta-learning (cf. Sutton, 1992; Bengio, 2000; Finn et al., 2017; Xu et al., 2018) to adapt the discount factor. The meta-gradient algorithm by Xu et al. (2018) uses the insight that the updates in Equations (2) and (4) are differentiable functions of hyper-parameters such as the discount. On the next sample or rollout of experience, using updated parameters $\mathbf{w} + \Delta\mathbf{w}(\gamma)$, written here as an explicit function of the discount, the agent then applies a gradient based actor-critic update, not to parameters $\mathbf{w}$, but to the parameter $\boldsymbol{\theta}$ that defines the discount $\gamma$ which is used in a standard learning update. Xu et al. (2018) demonstrated that this improved performance on Atari, while using a separate hand-tuned discount factor for the meta-update. We instead use the undiscounted returns ($\gamma_m = 1$) to define the meta-gradient updates, to understand if this technique can fully replace the need to reason about timescales and discounts.

A related heuristic, quite specific to Atari, is to track the number of lives that the agent has available (in several Atari games the agent is allowed to die a fixed number of times before the game is over), and hard code an episode termination ($\gamma = 0$) when this happens. We ignore the number of lives channel exposed by the Arcade Learning Environment in all our experiments.

## 2.2 Sculpting the agent-environment interface

In reinforcement learning we commonly assume that time progresses in discrete steps with a fixed duration. While algorithms are typically defined in this native space, learning at the fastest timescale provided by the environment is sometimes not the most practical or efficient solution, at least with the current generation of learning algorithms. It is often convenient to have the agent operate at a slower timescale, for instance by repeating each selected action a fixed number of times. The use of such fixed action repetitions is a widely used heuristic (e.g., Mnih et al., 2015; van Hasselt et al., 2016; Wang et al., 2016; Mnih et al., 2016) with several advantages. First, operating at a slower timescale increases the action gap (Farahmand, 2011), which can lead to more stable learning (Bellemare et al., 2015) because it becomes easier to appropriately rank actions reliably when the value estimates are uncertain or noisy. Second, selecting an action every few steps can save a significant amount of computation. Finally, committing longer to each action may help exploration, because the diameter of the solution space has effectively been reduced, for instance removing some often-irrelevant sequences of actions that jitter back and forth at a fast time scale.

A more general solution approach is for the agent to learn the most appropriate time scale at which to operate. Solving this problem in full generality is one of the aims of hierarchical reinforcement learning (Dayan & Hinton, 1993; Wiering & Schmidhuber, 1997; Sutton et al., 1998; Bacon et al., 2017). This general problem remains largely unsolved. A simpler, though more limited, approach is to allow the agent to learn how long to commit to each selected action (Lakshminarayanan et al., 2017). At each step $t$, the agent may be allowed to select both an action $A_t$ and a commitment $C_t$, by sampling from two separate policies, both trained with policy gradient. Committing to an action for multiple steps raises the issue of how to handle intermediate observations without missing out on the potential computational savings. Conventional deep RL agents for Atari max-pool multiple image frames into a single observation. In our setting, the agent gets one image frame as an observation after each new action selection. The agent needs to learn to trade-off the benefits of action repetition (e.g., lower variance, more directed behaviour) with its disadvantages (e.g., not being able to revise its choices during as often, and missing potentially useful intermediate observations).

Many state-of-the-art RL agents use non-linear function approximators to represent values, policies, and states. Being able to learn flexible state representations was essential to capitalize on the successes of deep learning, and to scale reinforcement learning algorithms to visually complex domains (Mnih et al., 2015). While the use of deep neural network to approximate value functions and policies is widespread, their input is often not the raw observations but the result of domain-specific heuristic transformations. In Atari, for instance, most agents rely on down-sampling the observations to an $84 \times 84$ grid (down from the original $210 \times 160$ resolution), grey scaling them, and finally concatenating them into a *K-Markov* representation, with $K = 4$. We replace this specific preprocessing pipeline with a state representation learned end-to-end. We feed the RGB pixel observations at the native resolution of the Arcade Learning Environment into a convolutional network with 32 and 64 channels (in the first and second layer, respectively), both using $5 \times 5$ kernels with a stride of 5. The output is fed to a fully connected layer with 256 hidden units, and then to an LSTM recurrent network (Hochreiter & Schmidhuber, 1997) of the same size. The policy for selecting the action and its commitment is computed as logits coming from two separate linear outputs of the LSTM. The network must then integrate information over time to cope with any issues like the flickering of the screen that had motivated the standard heuristic pipeline used by deep RL agents on Atari.

## 3 Experiments

When designing an algorithm it is useful to keep in mind what properties we would like the algorithm to satisfy. If the aim is to design an algorithm that is *general*, in addition to standard metrics such as asymptotic performance and data efficiency, there are additional dimensions that are useful to consider. Consider the following questions. 1) Does the algorithm require careful reasoning to select an appropriate time horizon for decision making? Manually setting the farsightedness of agents is tricky without domain knowledge or tuning. 2) How robust is the algorithm to the scaling of rewards? Rewards can have arbitrary scales, that may change by orders of magnitudes during training, as the agent improves. 3) Can the agent use commitment (e.g. action repetitions, or options) to alleviate the difficulty of learning at the fastest time scale? 4) Does the algorithm scale to complex domains? 5) How much tuning does the algorithm require to work on a new domain? Tuning is

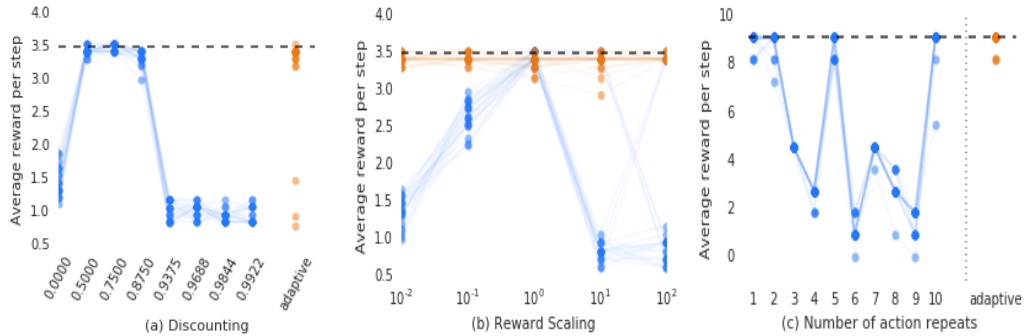

Figure 1: Investigations on the robustness of an A2C agent with respect to discounting, reward scaling and action repetitions. We report the average reward per environment step, after 5000 steps of training, for each of 20 distinct seeds. Each parameter study compares different fixed configurations of a specific hyper-parameter to the corresponding adaptive solution. In all cases the performance of the adaptive solutions is competitive with that of the best tuned solution

expensive, and combinatorially so when there are many different hyper-parameters because often the hyper-parameters interact in subtle ways. None of these dimensions is binary, and different algorithms may satisfy each of them to a different degree, but keeping them in mind can be helpful to drive research towards more general RL solutions. We first discuss the first three in isolation, in the context of simple toy environments, to increase the understanding about how adaptive solutions compare to the corresponding heuristics they are intended to replace. We then use the 57 Atari games in the Arcade Learning Environment (Bellemare et al., 2013) to evaluate the performance of the different methods at scale. Finally, we investigate how well the methods generalize to new domains, using 28 continuous control tasks in the DeepMind Control Suite (Tassa et al., 2018).

### 3.1 MOTIVATING EXAMPLES

We use a simple tabular actor-critic agent (A2C) to investigate in a minimal setup how domain heuristics and adaptive solutions compare with respect to some of these dimensions. We report average reward per step, after 5000 environment steps, for each of 20 replicas of each agent.

First, we investigate the role of discounting for effective learning. Consider a small chain environment with $T = 9$ states and 2 actions. The agent starts every episode in the middle of the chain. Moving left provides a -1 penalty. Moving right provides a reward of $2d/T$, where $d$ is the distance from the left end. When either end of the chain is reached, the episode ends, with an additional reward $T$ on the far right end. Figure 1a shows a parameter study over a range of values for the discount factor. We found the best performance was between 0.5 and 0.9, where learning is quite effective, but observed decreased performance for lower or higher discounts. This shows that it can be difficult to set a suitable discount factor, and that the naive solution of just optimizing the undiscounted return may also perform poorly. Compare this to the same agent, but equipped with the adaptive meta-gradient algorithm discussed in Section 2.1 (in orange in Figure 1.a). Even initializing the discount to the value of 0.95 (which performed poorly in the parameter study), the agent learned to reduce the discount and performed in par with the best tuned fixed discount.

Next, we investigate the impact of reward scaling. We used the same domain, but keep the discount fixed to a value of 0.8 (as it was previously found to work well). We examine instead the performance of the agent when all rewards are scaled by a constant factor. Note that in the plots we report the unscaled rewards to make the results interpretable. Figure 1.b shows that the performance of the vanilla A2C agent (in blue) degraded rapidly when the scale of the rewards was significantly smaller or larger than 1. Compare this to the same agent equipped with PopArt, we observe better performance across multiple orders of magnitude for the reward scales. In this tiny problem, learning could also be achieved by tuning the learning rate for each reward scale, but that does not suffice for larger problems. Adaptive optimization algorithm such as Adam (Kingma & Ba, 2014) or RMSProp

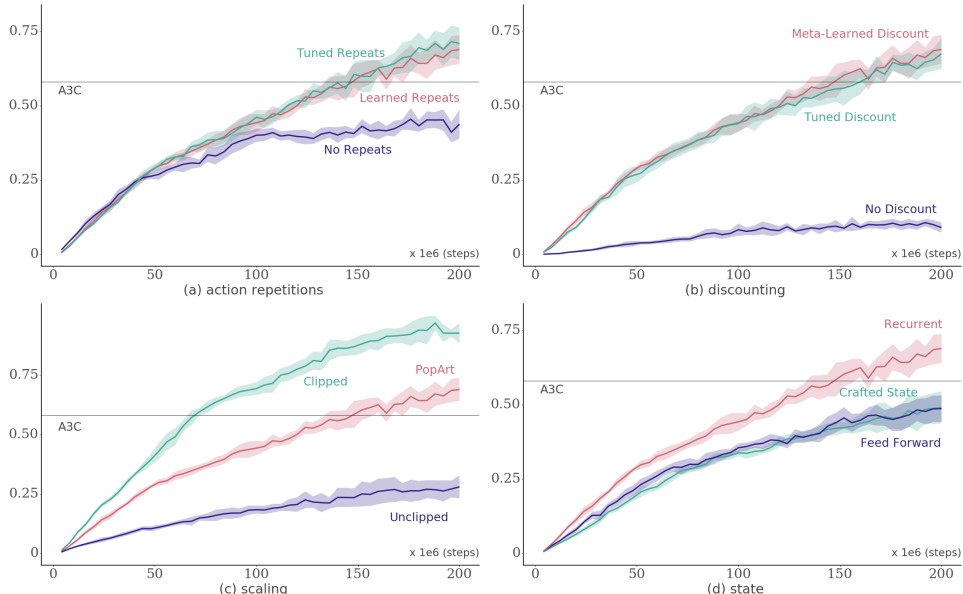

Figure 2: Comparison of inductive biases to RL solutions. All curves show mean episode return as a function of the number of environment steps. Each plot compares the same fully general agent to 2 alternative. a) tuned action repeats, and no action repeats. b) tuned discount factor, and no discounting. c) reward clipping, and learning from raw rewards with no rescaling of the updates. d) learning from the raw observation stream of Atari, and the standard preprocessing.

(Tieleman & Hinton, 2012) can also provide some degree of invariance, but, as we will see in Section 3.2, they are not as effective as PopArt normalization.

Finally, we investigate the role of action repeats. We consider states arranged into a simple cycle of 11 states. The agent starts in state 0, and only moves in one direction using one action, the other action does not move the agent. The reward is 0 everywhere, except if the agent selects the non-moving action in the $11 - th$ state: in this case the agent receives a reward of 100 and the episode ends. We compare an A2C agent that learns to choose the number of action repeats (up to 10), to an agent that used a fixed number of repetitions C. Figure 1.c shows how the number of fixed action repeats used by the agent is a sensitive hyper-parameter in this domain. Compare this to the adaptive agent that learns how often to repeat actions via policy gradient (in orange in Figure 1.c). This agent quickly learned a suitable number of action repeats and thereby performed very well. This is a general problem, in many domains of interest it can be useful to combine fine-grained control in certain states, with more coarse and directed behaviour in other parts of the state space.

## 3.2 PERFORMANCE ON LARGE DOMAINS

To evaluate the performance of the different methods on larger problems, we use A2C agents on many Atari games. However, differently from the previous experiments, the agent learns in parallel from multiple copies of the environment, similarly to many state-of-the-art algorithms for reinforcement learning. This configuration increases the throughput of acting and learning, and speeds up the experiments. In parallel learning training setups, the learning updates may be applied synchronously (Espeholt et al., 2018) or asynchronously (Mnih et al., 2016). Our learning updates are synchronous: the agent's policy takes steps in parallel across 16 copies of the environment to create multi-step learning updates, batched together to compute a single update to the parameters. We train individual agents on each game. Per-game scores are averaged over 8 seeds, and we then track the median human normalized score across all games. All hyper-parameters for our A2C agents were selected for a generic A2C agent on Atari before the following experiments were performed, with details given in the appendix.

Our experiments measure the performance of a full adaptive A2C agent with learned action repeats, PopArt normalization, learned discount factors, and an LSTM-based state representation. We compare the performance of this agent to agents with exactly one adaptive component disabled and replaced with one of two fixed components. This fixed component is either falling back to the environment specified task (e.g. learning directly from undiscounted returns), or using the corresponding fixed heuristic from DQN. These comparisons enable us to investigate how important the original heuristic is for current RL algorithms, as well as how fully an adaptive solution can replace it.

In the top left quadrant of Figure 2, we investigate action repeats and their impact on learning. We compare the fully general agent to an agent that used exactly 4 action repetitions (as tuned for Atari (Mnih et al., 2015)), and to an agent that acted and learned at the native frame rate of the environment. The adaptively learned solution performed almost as well as the tuned domain heuristic of always repeating each action 4 times. Interestingly, in the first 100M frames, also acting at the fastest rate was competitive with the agents equipped with action repetition (whether fixed or learned), at least in terms of data efficiency. However, while the agents with action repeats were still improving performance until the very end of the training budget, the agent acting at the fastest timescale appeared to plateau earlier. This performance plateau is observed in multiple games (see appendix), and we speculate that the use of multiple action repetitions may be helping achieve better exploration. It is worth noting that in wall-clock time the gap in the performance of the agents with action repetitions was even larger due to the cost of the additional compute.

In the top right quadrant, we investigate discounting. The agent that used undiscounted returns directly in the updates to policy and values performed very poorly, demonstrating that in complex environments the naive solution of directly optimizing the real objective is problematic with modern deep RL agents. Interestingly, while performance was very poor overall, the agent did demonstrate good performance on a few specific games. For instance, in `bowling` it achieved a better score than state of the art agents such as Rainbow (Hessel et al., 2018a) and ApeX (Horgan et al., 2018). The agent with tuned discount and the agent with a discount factor learned through meta-gradient RL performed much better overall. The adaptive solution did slightly better than the heuristic.

In the bottom left quadrant, we investigate the effect of reward scales. We compare the fully adaptive agent to an agent where clipping was used in place of PopArt, and to a naive agent that used the environment reward directly. Again, the naive solution performed very poorly, compared to using either the domain heuristic or the learned solution. Note that the naive solution is using RMSProp as an optimizer, in combination with gradient clipping by norm (Pascanu et al., 2012); together these techniques should provide at least some robustness to scaling issues, but in our experiments PopArt provided an additional large increase in performance. In this case, the domain heuristic (reward clipping) retained a significant edge over the adaptive solution. This suggests that reward clipping might not be helping exclusively with reward scales; the inductive bias of optimizing for a weighted frequency of rewards is a very good heuristic in many Atari games, and the qualitative behaviour resulting from optimizing the proxy objective might result in a better learning dynamics. While clipping was better in aggregate, PopArt yielded significantly improved scores on several games (e.g., `centipede`) where the clipped agent was stuck in sub-optimal policies.

Finally, we compare the fully end to end pipeline with a recurrent network, to a feedforward neural network with the standard Atari pipeline. The recurrent end to end solution performed best, showing that a recurrent network is sufficiently flexible to learn on its own to integrate relevant information over time, despite the Atari-specific features of the observation stream (such as the flickering of the screen) that motivated the more common heuristic approach.

## 3.3 GENERALIZATION TO NEW DOMAINS

Our previous analysis shows that learned solutions are mostly quite competitive with the domain heuristics on Atari, but do not uniformly provide additional benefits compared to the well tuned inductive biases that are commonly used in Atari. To investigate the generality of these different RL solutions, in this section we compare the fully general agent to an agent with all the usual inductive biases, but this time we evaluate them on a completely different benchmark: a collection of 28 continuous control tasks in the DeepMind Control Suite. The tasks represent a wide variety of physical control problems, and the dimension of the real-valued observation and action vectors

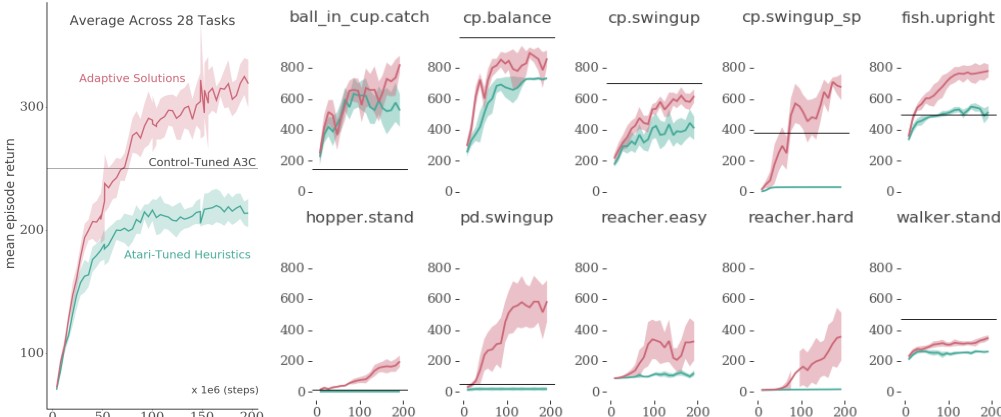

Figure 3: In a separate experiment on the 28 tasks in the DeepMind Control Suite, we compared the general solution agent with an A2C agent using all the domain heuristics previously discussed. Both agents were trained and evaluated on the new domain with no changes to the algorithm nor any additional tuning for this very different set of environments. On average, the general adaptive solutions transfer better to the new domain that the heuristic solution. On the left we plot the average performance across all 28 tasks. On the right we show the learning curves on a selection of 10 tasks.

differs across the tasks. The environment state can be recovered from the observation in all but one task. The rewards are bounded between 0 and 1, and tasks are undiscounted.

Again we use a parallel A2C implementation, with 16 copies of the environment, and we aggregate results by first averaging scores across 8 seeds, and then taking the mean across all 28 tasks. Because all tasks in this benchmark are designed to have episode returns of comparable magnitude, there is no need to normalize the results to meaningfully aggregate them. For both agents we use the exact same solutions that were used in Atari, with no additional tuning. The agents naturally transfer to this new domain with two modifications: 1) we do not use convolutions since the observations do not have spatial structure. 2) the outputs of the policy head are interpreted as encoding the mean and variance of a Gaussian distribution instead of as the logits of a categorical one.

Figure 3 shows the fully general agent performed much better than the heuristic solution, which suggests that the set of inductive biases typically used by Deep RL agents on Atari do not generalize as well to this new domain as the set of adaptive solutions considered in this paper. This highlights the importance of being aware of the priors that we incorporate into our learning algorithms, and their impact on the generality of our agents. On the right side of Figure 3, we report the learning curves on the 10 tasks for which the absolute difference between the performance of the two agents was greatest (details on the full set of 28 tasks can be found in Appendix). The adaptive solutions performed equal or better than the heuristics on each of these 10 tasks, and the results in the appendix show performance was rarely worse. The reference horizontal black lines mark the performance of an A3C agent, tuned specifically for this suite of tasks, as reported by Tassa et al. (2018). The adaptive solution was also better, in aggregate, than this well tuned baseline; note however the tuned A3C agent achieved higher performance on a few games.

## 4   RELATED WORK AND DISCUSSION

The present work was partially inspired by the work of Silver et al. (2017) in the context of Go. They demonstrated that specific domain specific heuristics (e.g. pretraining on human data, the use of handcrafted Go-specific features, and exploitation of certain symmetries in state space), while originally introduced to simplify learning (Silver et al., 2016), had actually outlived their useful-ness: taking a *purer* approach, even stronger Go agents could be trained. Importantly, they showed removing these domain heuristics, the same algorithm could master other games, such as Shogi and Chess. In our paper, we adopted a similar philosophy but investigated the very different set of domain specific heuristics, that are used in more traditional deep reinforcement learning agents.

Our work relates to a broader debate (Marcus, 2018) about priors and innateness. There is evidence that we, as humans, posses specific types of biases, and that these have a role in enabling efficient learning (Spelke & Kinzler, 2007; Dubey et al., 2018); however, it is far from clear the extent to which these are essential for intelligent behaviour to arise, what form these priors take, and their impact on the generality of the resulting solutions. In this paper, we demonstrate that several heuristics we commonly use in our algorithms are already harming the generality of our methods. This does not mean that other different inductive biases could not be useful as we progress towards more flexible, intelligent agents; it is however a reminder that we must be careful with the domain knowledge and priors we bake into our solutions, and we must be ready to revise them over time.

We found that existing learned solutions are competitive with well tuned domain heuristics, even on the domain these heuristics were designed for, and they seem to generalize better to unseen domain. This makes a case for removing these biases in future research on Atari, since they are not essential for competitive performance, and they might hide issues in the core learning algorithm. The only case where we still found a significant gap in favour of the domain heuristic was in the case of clipping. While, to the best of our knowledge, PopArt does address scaling issues effectively, clipping still seems to help on several games. Changing the reward distribution has many subtle implications for the learning dynamics, beside affecting the magnitude of updates (e.g. exploration, risk-propensity, ...). We believe it is a fruitful direction for future research to investigate what other general solutions could be deployed in our agents to fully recover the advantages of clipping.

Several of the biases that we considered have *knobs* that could be tuned rather than learned (e.g. the discount, the number of repeats, etc); however, this is not a satisfying solution for several reasons. Tuning is expensive, and these heuristics interact subtly with each other, thus requiring to explore a combinatorial space to find suitable settings. Consider the use of fixed action repeats: when changing the number of repetitions you also need to change the discount factor, otherwise this will change the effective horizon of the agent, which in turn affects the magnitude of the returns and therefore the learning rate. Also, a fixed tuned value might still not give you the full benefits of an adaptive learned approach that can adapt to the various phases of the training process.

There are other two features of our algorithm that, despite not incorporating quite as much domain knowledge as the heuristics discussed in this paper, also constitute a potential impediment to its generality and scalability. 1) the use of parallel environments is not always feasible in practice, especially in real world applications (although recent work on robot farms Levine et al. (2016) shows that it might still be a valid approach when sufficient resources are available). 2) the use of back-propagation through time for training recurrent state representations constrains the length of the temporal relationship that we can learn, since the memory consumption is linear in the length of the rollouts. Further work in overcoming these limitations, successfully learning online from a single stream of experience, is a fruitful direction for future research.

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

## 5 TRAINING DETAILS

We performed very limited tuning on Atari, both due to the cost of running so many comparison with 8 seeds at scale across 57 games, and because we were interested in generalization to a different domain. We used a learning rate of $1e-3$, an entropy cost of $0.01$ and a baseline cost of $0.5$. The learning rate was selected among $1e-3$, $1e-4$, $1e-5$ in an early version of the agent without any of the adaptive solutions, and verified to be reasonable as we were adding more components. Similarly the entropy cost was selected between $0.1$ and $0.01$. The baseline cost was not tuned, but we used the value of $0.5$ that is common in the literature. We used the TensorFlow default settings for the additional parameters of the RMSProp optimizer.

The learning updates were batched across rollouts of 150 agent steps for 16 parallel copies of the environment. All experiments used gradient clipping by norm, with a maximum norm of 5., as in Wang et al. (2016). The adaptive agent could choose to repeat each action up to 6 times. PopArt used a step-size of $3e-4$, with bounds on the scale of $1e-4$ and $1e6$ for numerical stability, as reported by Hessel et al. (2018b). We always used the undiscounted return to compute the meta-gradient updates, and when using the adaptive solutions these would update both the discount $\gamma$ and the trace $\lambda$, as in the experiments by Xu et al. (2018). The meta-updates were computed on smaller rollouts of 15 agent steps, with a meta-learning rate of $1e-3$. No additional tuning was performed for any of the experiments on the Control Suite.

## 6 EXPERIMENT DETAILS

In Figure 4 we report the detailed learning curves for all Atari games for three distinct agents: the fully adaptive agent (in red), the agent with fixed action repeats (in green), and the agent acting at the fastest timescale (in blue). It's interesting to observe how on a large number of games (`ice_hockey`, `jamesbond`, `robotank`, `fishing`, `double_dunk`, etc) the agent with no action repeats seems to learn stably and effectively up to a point, but then plateaus quite abruptly. This suggests that exploration might be a major reason for the reduced performance of this agent that we observed in aggregate across the benchmark in the later stages of training.

In Figure 5 and 6 we show the discount factors and the eligibility traces $\lambda$ as they were meta-learned over the course of training by the fully adaptive agent. We plot the soft time horizons $T = (1-\gamma)^{-1}$ and $T = (1-\lambda)^{-1}$. It's interesting to observe how diverse these are for the various games. Consider for instance the discount factor $\gamma$: in games such as `robotank`, `bank_heist`, and `tutankham` the horizon increases up to almost 1000, or even above, while in other games, such as `surround` and `double_dunk` the discount reduces over time. Also for the eligibility trace parameter $\lambda$ we observe very diverse values in different games. In Figure 7 we report the complete set of learning curves for all 28 tasks in the Control Suite (the 10 selected for the main text are those for which the difference in performance between the fully adaptive agent and the heuristic agent is greater).

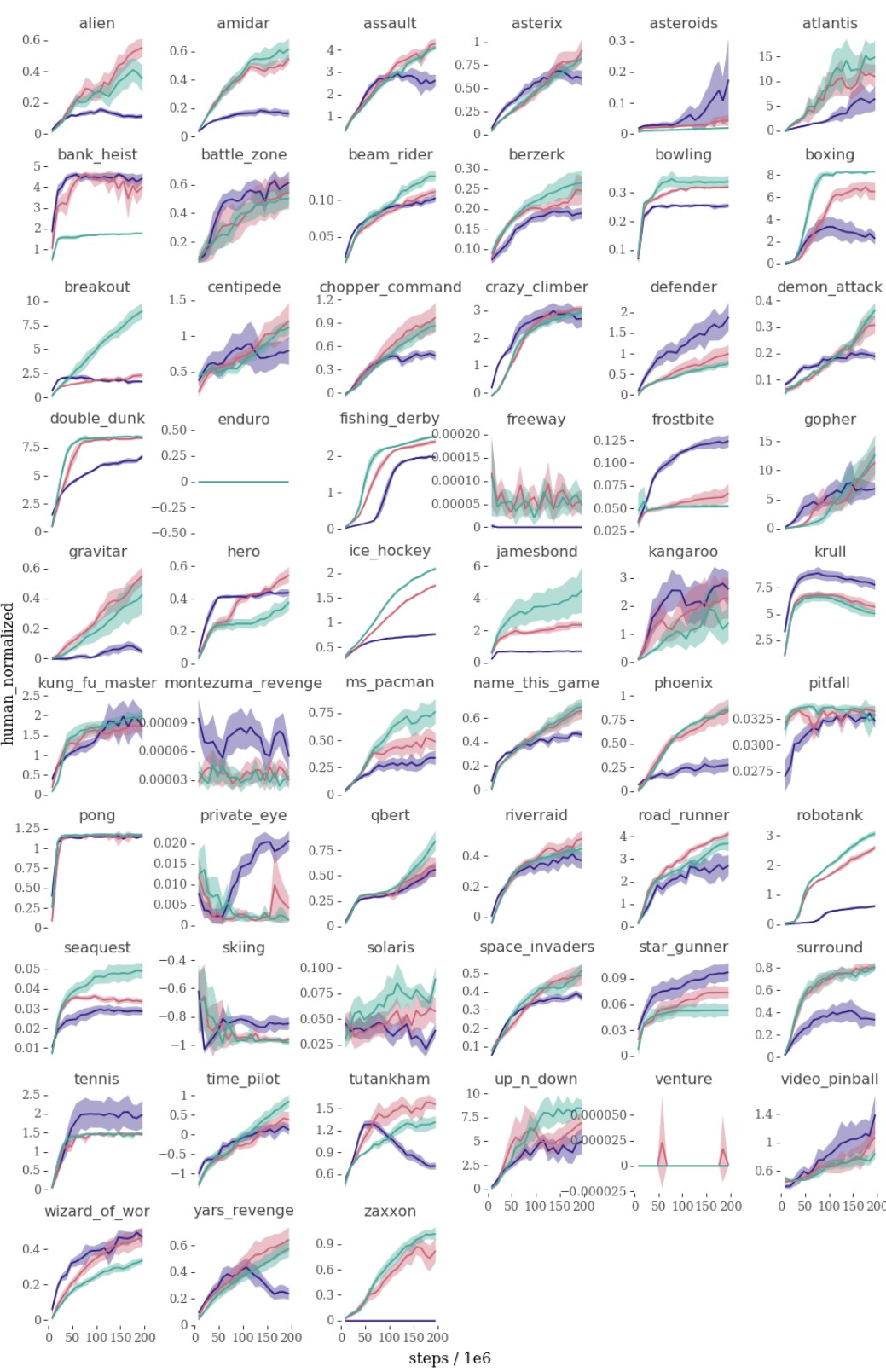

Figure 4: Performance on all Atari games of the fully adaptive agent (in red), the agent with fixed action repeats (in green), and the agent acting at the fastest timescale (in blue).

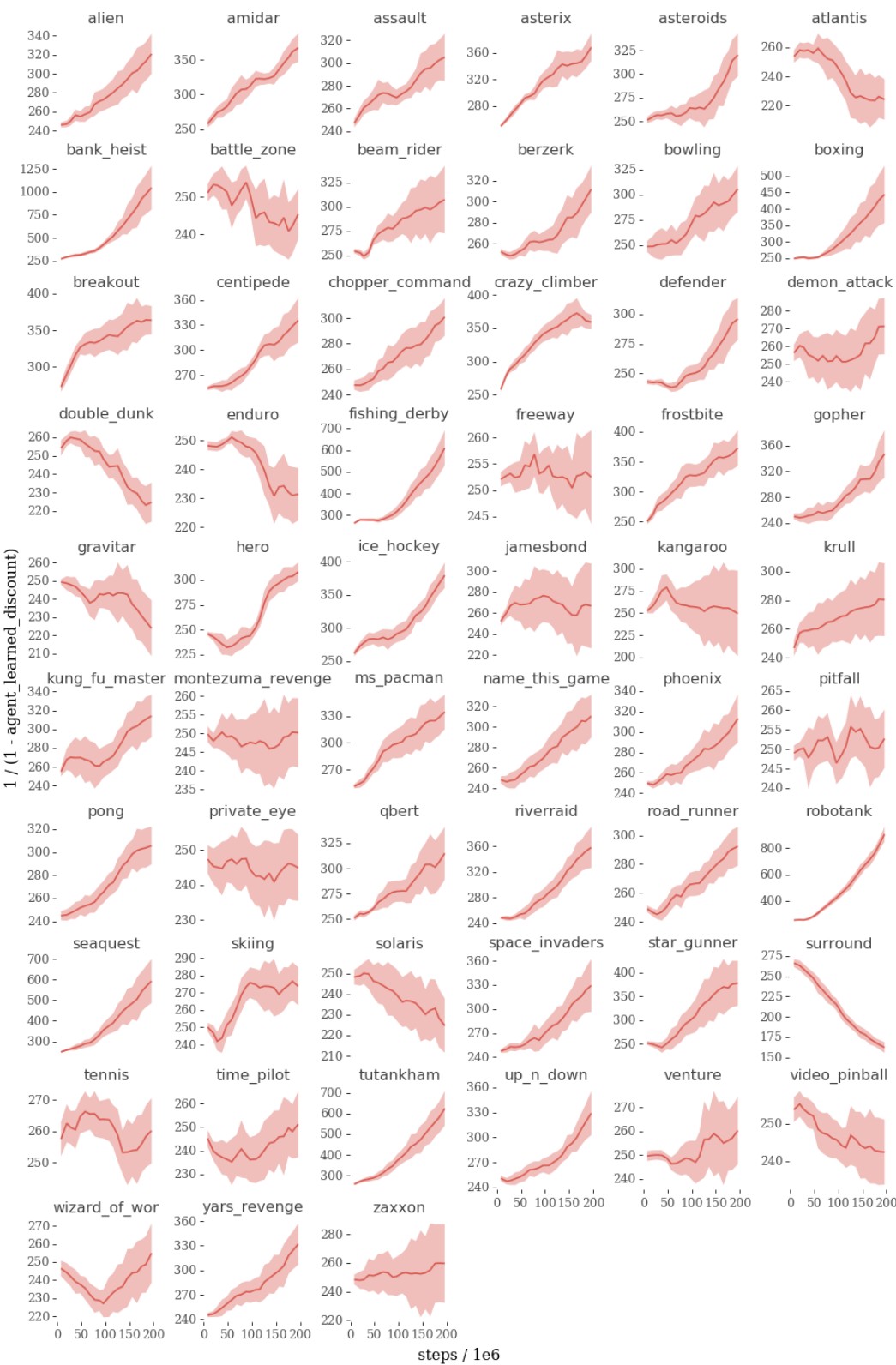

Figure 5: For each Atari game, the associated plot shows the time horizon $T = (1 - \gamma)^{-1}$ for the discount factor $\gamma$ that was adapted across the 200 million training frames.

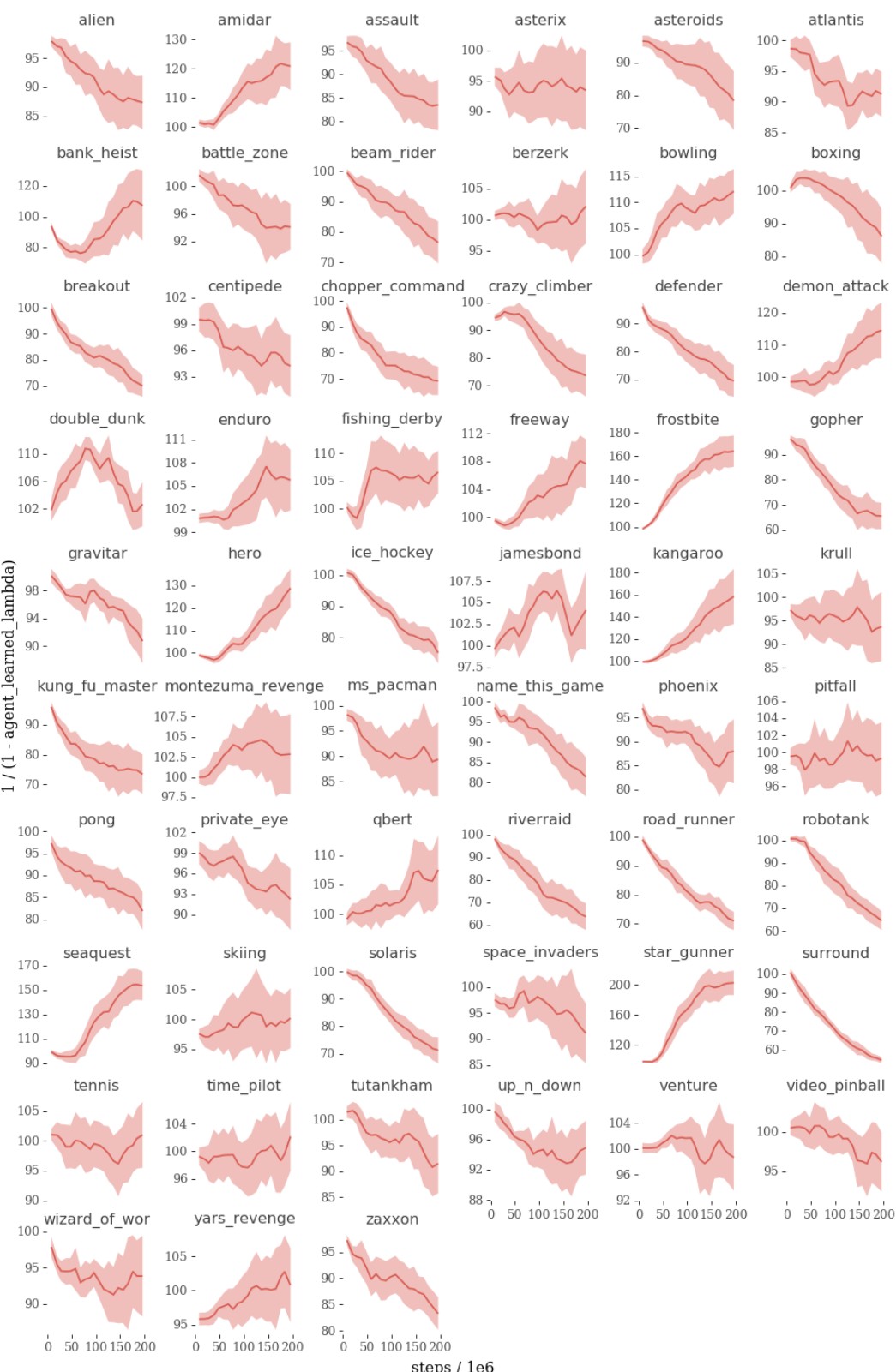

Figure 6: For each Atari game, the associated plot shows the time horizon $T = (1 - \lambda)^{-1}$ for the eligibility trace $\lambda$ that was adapted across the 200 million training frames.

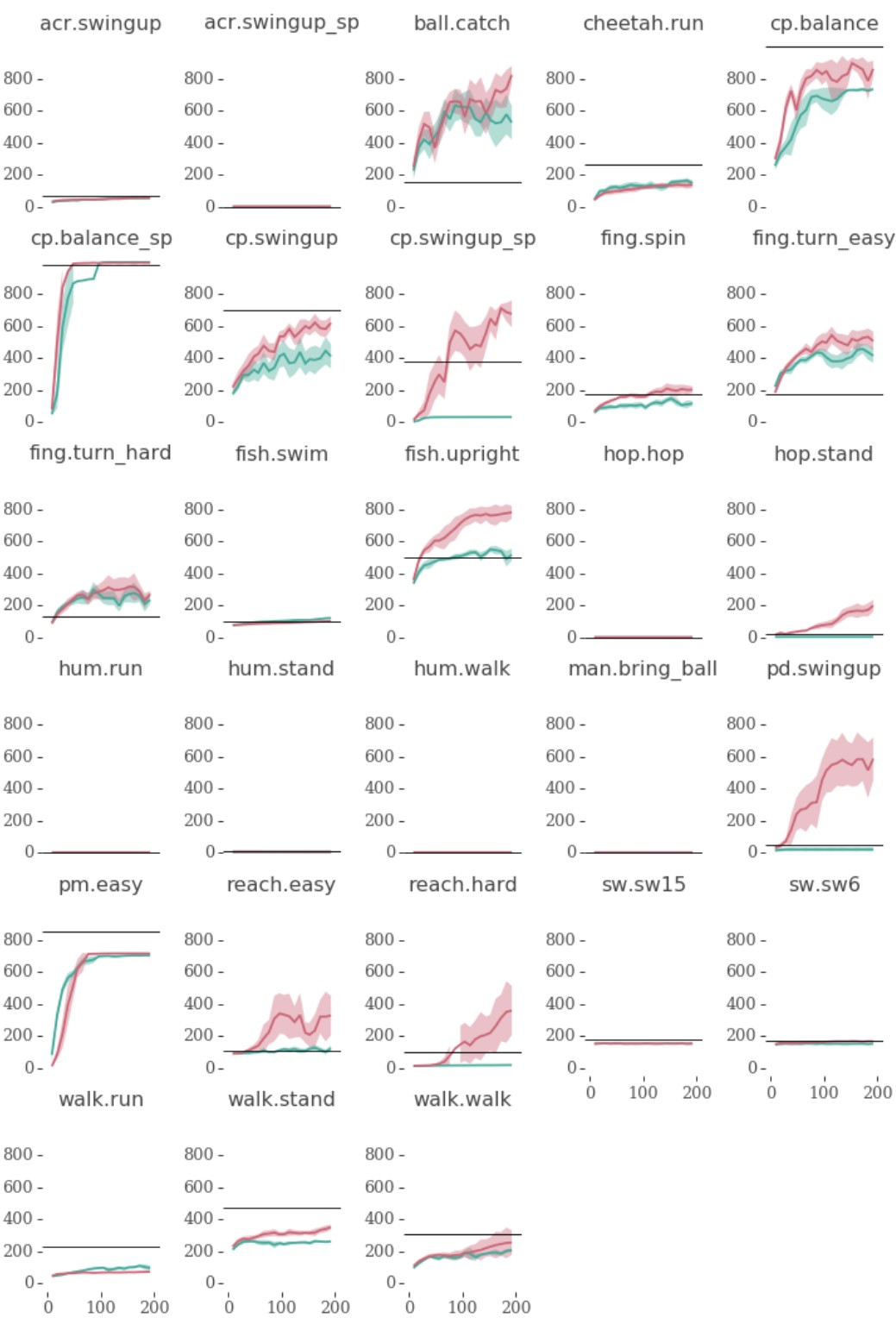

Figure 7: Performance on all control suite tasks for the fully adaptive agent (in red), the agent trained with the Atari heuristics (in green), and an A3C agent (black horizontal line) as tuned specifically for the Control Suite by Tassa et al. (2018).

