# OpenReview forum: "On Inductive Biases in Deep Reinforcement Learning"
_ICLR.cc/2019/Conference_

### Official Review · AnonReviewer1 · 2018-11-02
**Good summary and experimental evaluation of various inducive biases in deep reinforcement learning**

**Rating:** 7
**Confidence:** 2

**Review:**

The paper presents and evaluates different common inductive biases in Deep RL. These are systematically evaluated on different experimental settings.

The paper is easy to read and the authors explain well the setting and their findings. The comparison and evaluations is well conducted and valuable contribution to the literature.  I would have liked some more details on the motivating example in section 3.1, maybe with a figure supporting the explanation of the example.

---

> ### Author Response · Authors · 2018-11-16
> **Thanks for the comments and suggestions**
>
> We thank the reviewer for the many positive comments.
> We will add a figure for each of the 3 motivating examples in the Appendix, thanks for the suggestion!

---

### Official Review · AnonReviewer2 · 2018-11-04
**This paper contains various numerical experiments to see the effects of some heuristics in reinforcement learning, but no definite answers are given.**

**Rating:** 3
**Confidence:** 4

**Review:**

This paper contains various numerical experiments to see the effects of some heuristics in reinforcement learning. Those heuristics include reward clipping, discounting for effective learning, repeating actions, and different network structures. However, since the training algorithms also greatly affect the performance of RL agents, it seems hard to draw any quantitive conclusions from this paper.

Detailed comments:

1. It seems that actor-critic algorithms are defined for RL with function approximation. What is the tabular A2C algorithm? A reference in Section 3.1 would be better.

2. This paper claims to study the "inductive biases", which is not clearly defined. How to quantify those biases and how to measure "generality"?

3. Are there any quantitive conclusions that can be drawn from the experiments?

4. Since the performance of RL agents also relies on initialization and the training algorithms. There are a lot of tricks of optimization for deep learning. How to measure the "inductive biases" by ruling out the effects of training algorithms?

---

> ### Author Response · Authors · 2018-11-16
> **On "inductive biases", "generality" and other questions.**
>
> Some of the questions raised by the reviewer suggest that there may have been a misunderstanding of the term “inductive bias”, possibly interpreted as referring to some form of statistical bias. “Inductive Bias” is a well defined concept from the Machine Learning and Neuroscience literature and refers to the set of assumptions that go into a learning system (such as domain knowledge and heuristics). In the context of this paper we define and classify the various types of inductive biases under consideration in Section 2.
>
> Regarding how to measure "generality": in this paper we propose to measure the "generality" of an RL algorithm as the degree to which such algorithm can be ported to a different domain from the one it was proposed for, without forcing the practitioner to revisit the inductive biases that were incorporated in the original agent. Our experiments on Continuous Control show that adaptive solutions perform better in this respect than other heuristic inductive biases.
>
> As always, the Actor-Critic update in equation 2 of Section1 subsumes the tabular case, which can be seen by noting that in a tabular representation the gradient would only update the corresponding entry in the table.

---

### Official Review · AnonReviewer3 · 2018-11-05
**Review for the paper: "On Inductive Biases in Deep Reinforcement Learning"**

**Rating:** 3
**Confidence:** 4

**Review:**

This paper focuses on deep reinforcement learning methods and discusses the presence of inductive biases in the existingRL algorithm. Specifically, they discuss biases that take the form of domain knowledge or hyper-parameter tuning. The authors state that such biases rise the tradeoff between generality and performance wherein strong biases can lead to efficient performance but deteriorate generalization across domains. Further, it motivates that most inductive biases has a cost associated to it and hence it is important to study and analyze the effect of such biases.

To support their insights, the authors investigate the performance of well known actor-critic model in the Atari environment after replacing domain specific heuristics with the adaptive components. The author considers two ways of injecting biases: i) sculpting agents objective and ii) sculpting agent's environment. They show empirical evidence that replacing carefully designed heuristics to induce biases with more adaptive counterparts preserves performance and generalizes without additional fine tuning.

The paper focuses on an important concept and problem of inductive biases in deep reinforcement learning techniques.
Analysis of such biases and methods to use them judiciously is an interesting future direction. The paper covers a lot of related work in terms of various algorithms and corresponding biases.
However, this paper only discusses such concepts at high level and provides short empirical evidences in a single environment to support their arguments. Further, both the heuristics used in practice and the adaptive counterparts that the paper uses to replace those heuristics are all available in existing approaches and there is no novel contribution in that direction too.
Finally, the adaptive methods based on parallel environment and RNNs have several limitation, as per author's own admission.

Overall, the paper does not have any novel technical contributions or theoretical analysis on the effect of such inductive biases which makes it very weak. Further, there is nothing surprising about the author's claims and many of the outcomes from the analysis are expected. The authors are recommended to consider this task more rigorously and provide stronger and concrete analysis on the effects of inductive biases on variety of algorithms and variety of environments.

---

> ### Author Response · Authors · 2018-11-16
> **On the empirical evidence provided in the paper**
>
> In addition to 3 grid-world domains (designed specifically to highlight specific properties of the inductive biases considered in the paper), we also provide extensive experiments at scale on 57 Atari games and 28 continuous control tasks. This is a larger set of non-trivial environments than in the vast majority of deep RL papers. Perhaps the reviewer interpreted the Atari experiments (on 57 games) as having been run on a single Atari game?

---

> > ### Comment · AnonReviewer3 · 2018-11-27
> > **Did not misunderstand but also not a big factor in review**
> >
> > Thanks for clarifying question: I did not misunderstand the empirical efforts. As you can see, my review mentions single "environment". I believe you conduct 57 games experiment in Arcade environment. Perhaps when I used Atari environment, it confused you.
> >
> > In any case, the major issue of no substantial technical contributions and/or theoretical analysis still stands and hence my score also stands. With these two ingredients, the current empirical evaluation may (not necessarily) have been adequate but the paper needs more work in terms of contributions before it can be accepted.

---

### Meta-Review · Area_Chair1 · 2018-12-15
**Not enough novel technical content nor insights**

**Confidence:** 4
**Recommendation:** Reject

**Metareview:**

The paper studies inductive biases in DRL, by comparing with different reward shaping, and curriculums. The authors performed comparative experiments where they replace domain specific heuristics by such adaptive components.

The paper includes very little (new) scientific contributions, and, as such, is not suitable for publication at ICLR.